# Intercepted Mosquitoes at New Zealand’s Ports of Entry, 2001 to 2018: Current Status and Future Concerns

**DOI:** 10.3390/tropicalmed4030101

**Published:** 2019-07-05

**Authors:** Sherif E. Ammar, Mary Mclntyre, Tom Swan, Julia Kasper, José G. B. Derraik, Michael G. Baker, Simon Hales

**Affiliations:** 1Department of Public Health, University of Otago, Wellington 6021, New Zealand; 2Australian Institute of Tropical Health and Medicine, James Cook University, Queensland 4814, Australia; 3Museum of New Zealand, Te Papa Tongarewa, Wellington 6011, New Zealand; 4Liggins Institute, University of Auckland, Auckland 1142, New Zealand

**Keywords:** mosquitoes, New Zealand, interception, *Aedes albopictus*, *Aedes aegypti*, used tyres, machinery, climate change, vector-borne diseases

## Abstract

Mosquito vectors are extending their range via international travel and trade. Climate change makes New Zealand an increasingly suitable environment for less tropically adapted exotic mosquito vectors to become established. This shift will add a multiplier effect to existing risks of both the establishment of new species and of resident exotic species extending into new areas. We describe trends in the border interceptions of exotic mosquitoes and evaluate the role of imported goods as a pathway for these introductions. *Ae. aegypti* and *Ae. albopictus*, the two most commonly intercepted species, were only intercepted in Auckland. Used tyres and machinery were the main mode of entry for both species. The majority of *Ae. albopictus* were transported as larvae by sea, while most *Ae. aegypti* were transported as adults by air. Continuing introductions of these mosquitoes, mainly arriving via Japan or Australia, increase the risk of the local transmission of mosquito-borne diseases in New Zealand in general and in the Auckland region in particular. These findings reinforce the need for a high performing and adequately resourced national biosecurity system, particularly port surveillance and inspection. Recommended biosecurity improvements are described.

## 1. Introduction

The rate of introduction of exotic mosquito species in new geographic areas has increased notably in parallel with global trade and travel [1,2,3,4]. The spread of the container-breeding species *Aedes albopictus* and *Aedes aegypti* beyond their native range has been facilitated by the trade of goods, significantly in used tyres and machinery [5,6,7]. These two species are a major public health concern, as they are the main vectors of the most important arboviral diseases, including yellow fever, dengue, chikungunya, and Zika [8]. Despite the widespread distribution of these arboviruses, the majority are found in tropical and subtropical climate zones where *Aedes* mosquitos are prevalent [9].

*Ae. albopictus* is native to humid tropics of Southeast Asia. However, it has expanded geographically during the last four decades [10,11] to mid-latitude temperate areas of all continents [12,13], making it the most invasive mosquito species in the world [13]. The main factor contributing to the aggressive colonizing capacity of *Ae. albopictus* seems to be its ability to adapt to different climates through the production of diapausing cold-resistant eggs, with temperate strains surviving cold winters in mid-latitudes [11,12].

*Ae. aegypti* is distributed predominantly in the tropical regions of all continents, as well as some subtropical regions, such as the south-eastern United States, northern Australia, and northern India [14]. Up to the beginning of the 20th century, this invasive species was established in temperate countries in Europe, as far north as Belarus and Ukraine [15]. It has recently re-established itself on the island of Madeira (south-west of Portugal) [16], around the Black Sea [17], and in Netherlands [18]. Unlike *Ae. albopictus*, *Ae. aegypti* does not overwinter in the egg stage. Instead, the active stages minimize exposure to unsuitable environmental conditions by utilizing sheltered sites in urban settings (e.g., water tanks) [19]. Reiter [15], however, claims that there are no climatic reasons why *Ae. aegypti* could not become widely established in other temperate regions, if introduced or reintroduced.

The transport of immature stages of *Aedes* species via the used tyre trade appears to account for the establishment of *Ae. japonicus* in France [20] and the United States [2,21]; *Ae. albopictus* in Albania, France, and Italy in Europe [12,22]; the United States, 10 other countries in the Americas [23]; and elsewhere [2,21,24]. The establishment of *Ae. albopictus* in the United States for the first time in 1985 followed a substantial increase in used tyre imports shipped from Japan after 1980. Subsequently, *Ae. albopictus* was recorded in 26 states [25]. By the late 1990s, the United States Centers for Disease Control and Prevention (CDC) stopped tyre inspections because *Ae. albopictus* had already invaded most of the country [26]. Used tyre transportation may not have played such a major role in recent invasions of *Ae. aegypti* [2]. However, used tyre shipments [12,22] from the United States were the major source of *Ae. aegypti* mosquitoes transported to South America [27] and the Netherlands [28].

There are also concerns over the introduction and establishment of exotic mosquito vectors by aircraft into countries where they are not indigenous [1]. Aircraft were probably responsible for the introduction of the *Ae. aegypti* into Trinidad and Tobago [29], Bermuda [30], Bolivia, and Colombia [1]. Further, dengue and Ross River Virus (RRV) infection outbreaks followed the introduction of *Ae. albopictus* by aircraft into the Solomon Islands [1], *Ae. vigilax* into Fiji [31], and most of the Western Pacific region [1,31]. 

The establishment of mosquitoes depends on suitable ecological conditions. Despite the predominantly temperate climate (subarctic in the north and subtropical in the south) [32], *Ae. albopictus* is widespread in Japan, with northward expansion (to latitude 38 degrees north; annual mean temperature of 11 °C) [33]. For the first time since World War II, Tokyo recorded an outbreak of dengue in early autumn 2014 [34]. Dengue epidemics in Japan are likely to increase over the next decades, facilitated by the continuing geographic expansion of *Ae. albopictus* and favorable climatic conditions [35]. Increasing temperature has been also implicated as a major factor in the establishment and re-establishment of *Aedes* species and their associated diseases in Europe [4,34].

Global climate change is projected to have a marked effect on larval development, female feeding behaviour, arbovirus replication, and transmission [36,37,38,39]. Global climate models project a rise in mean temperatures of 1.5 °C by sometime between 2030 and 2052 [40]. This change would create new ecological niches for mosquito vectors, altering the global spatio-temporal distribution of mosquito-borne diseases [41,42]. Consideration of adaptability and dispersal ability, combined with climate change projections and current risk mapping, suggest that new introductions and establishment of at least one of the two *Aedes* species are very likely to occur in new geographic areas [10,11] including those countries with rigorous biosecurity systems, such as France, Australia, and New Zealand [43].

New Zealand has only 12 documented native mosquito species, despite its temperate climate and suitable environments for mosquitoes to establish [44,45]. New Zealand’s peculiar indigenous fauna (notably a lack of land mammals as hosts), high level of anthropogenic environmental change [39], and increasing global trade (mainly shipping) and tourism make it vulnerable to invasion by exotic mosquitoes. All three resident mosquito species that are known vectors of human disease have been introduced: *Ae. notoscriptus, Ae. Australis*, and *Culex quinquefasciatus* [45]. Their establishment may serve as a blueprint for the establishment of other exotic mosquitoes in New Zealand, especially *Ae. albopictus* [46,47]. Laird et al. [46] first alerted authorities that used tyres were a source of imports and potential infestation by known vector species. He reported about a hundred *Ae. albopictus* larvae in a used tyre shipment from Japan. Derraik [48] found in 2006 that used tyres and machinery comprised about 75% of all mosquito interceptions arriving by ships in New Zealand. According to Kramer et al. [39], at least two endemic species (*Cx. pervigilans* and *Ae. antipodeus*) are also potential arbovirus vectors.

In this study, we describe trends in the border interceptions of exotic mosquitoes and evaluate the role of used tyre and vehicle imports as a means of transport. We update and expand on the review of Derraik [48] on mosquitoes intercepted in New Zealand to 2004. The study objectives are:To examine New Zealand exotic mosquito interception data, pathways, and ports of entry for the period from July 2001 to March 2018.To examine New Zealand import data for potential water receptacles (used tyres and used machinery) in the same period of time.To evaluate the role of used tyres and vehicles imports as a contributor to exotic mosquito introductions, especially for the container-breeding species, *Ae. albopictus* and *Ae. aegypti.*To examine interceptions of new vector mosquitoes as a risk factor for local transmission of arbovirus disease in New Zealand, and to consider implications of identified trends for present and projected climate conditions and for biosecurity practices.

## 2. Materials and Methods 

### 2.1. The Interception Records

Mosquito interception data were provided by the New Zealand Ministry of Health from records obtained between June 2001 and March 2018. The data were categorised according to mosquito species, country of origin, mode of entry, port of entry, and date of interception. The New Zealand Ministry of Health [49] defines an interception event as the confirmation that adult mosquitoes or larvae of public health significance are detected at or before the New Zealand border, or in association with recently arrived travellers or goods. Considering the impact of current biosecurity import practices on preventing the arrival of mosquitoes from overseas, we have adopted an even wider definition. We considered an interception event to be the detection of any mosquito, dead or alive, of foreign or unknown country of origin, irrespective of whether the species is already present in New Zealand (established). If more than one mosquito species were intercepted at the same time, each species (regardless of numbers of specimens) was considered as a separate interception event. Vessels, mainly ships, stop at one or more ports prior to arriving in New Zealand. This makes it difficult to confirm the origin of an invading species. We followed Derraik [48] in assigning origin to the last overseas port of call but appreciate that this does include an element of uncertainty. According to the Ministry of Health [50], the exotic mosquitoes listed on the “Unwanted Organisms Register” in New Zealand include all mosquitoes of the genus *Anopheles* plus 13 other species.

According to New Zealand BioSecure [45], there are three lines of defence to prevent the establishment of exotic mosquitoes in New Zealand. These lines are: (i) pre-border clearance of risk goods conducted by Ministry for Primary Industries Quarantine Service staff at offshore sites; (ii) inspection and disinsection, undertaken by public health units and port companies, of ships (first port of international call vessels), aircraft, and their high-risk cargo arriving at New Zealand ports; and (iii) mosquito surveillance at seaports and airports conducted by Public Health Units and the Ministry of Health, including handovers from Ministry for Primary Industries, from customs, or transitional facilities. 

The surveillance consists of routine monitoring surveys including adult (CO_2_ baited light traps) and larvae (World Health Organization standard tyre traps) trapping, larval surveys (World Health Organization standard dipping method) and interception responses. Most of the interception data were obtained from activities of the inspection and disinsection of ships and shipment.

### 2.2. Interception Sites: Airports, Seaports, and Their Transitional Facilities

Data were examined from seven airports, six seaports, and their transitional facilities (Figure 1). These are the major commercial ports of entry to New Zealand out of a total of 35 air and seaports where biosecurity surveillance is undertaken. The transitional facilities are approved to receive containers and goods that pose a potential biosecurity risk, especially plants, animals, and related products. At these facilities, the goods or containers are inspected or treated before they can be cleared for entry into the country [45]. There are about 7000 transitional facilities throughout New Zealand [51]. The entrance pathways of mosquitoes intercepted in transitional facilities were assigned to the nearest port and the means of invasion (e.g., fruit containers, used tyres, and used machinery) in relation to its origin of transport.

### 2.3. Trade Data Imports and International Flights

Data on air and sea freight imports (including used tyres and vehicles), and international passenger flight arrivals to New Zealand (2001 to 2017), were obtained from Statistics New Zealand. Used vehicles included motor cars, heavy vehicles, and all special purpose vehicles or transporters. Interception data were examined in relation to date, origin, and size of trade imports to assess the risk of exotic mosquitoes arriving in New Zealand and their pathways of entry.

### 2.4. Statistical Analysis

Confidence intervals for the proportions of interception records were estimated using the Clopper-Pearson exact method. The seasonality for *Ae. aegypti* and for *Ae. albopictus* interception records were estimated using the Poisson regression model of Stolwijk et al. [52]. Time trends in interception counts were modelled with negative binomial regression with overdispersion using the Genmod procedure in the statistical package SAS 9.4 (SAS Institute Inc., Cary, NC, USA). For all mosquito interception counts, a model with the year was used. For *Ae. albopictus* and *Ae. aegypti*, differences in time trends were tested with a model with the year, species, and species by year interaction terms.

## 3. Results

### 3.1. The Interception Records

**Interception events:** District Health Board officers and New Zealand BioSecure responded to over 650 suspected interceptions in the assessment period (Figure 2). Of these, 244 were considered interception events and used in the analysis. About 90% of these events had an identified foreign country of origin. However, more than 13.5% of the suspected interceptions were flies other than mosquitoes (mainly Chironomids). Of the 44 mosquito species intercepted, 18 were on the “Unwanted Organisms Register” (Table 1). These made up 75% of exotic interceptions. *Ae. aegypti* was the most commonly intercepted of the registered species, with more than 32% of the total, followed by *Ae. albopictus*, (22%). The number of *Ae. aegypti* and *Ae. albopictus* interceptions is likely to be higher, since 49 out of 244 records were taxonomically unidentifiable or identified to genus level only (Figure 2). The established species (*Ae. camptorhynchus*, *Cx. quinquefasciatus* and *Ae. notoscriptus*) of presumed foreign origin were intercepted 56 times. Although *Ae. camptorhynchus* was declared to be eradicated from New Zealand in June 2010, it remains a high-risk species on the register [53].

**Origin of interception:** Twenty-eight interceptions were of unknown origin (Figure 2). The balance of 179 interceptions came from 33 different countries (Table 2), of which Asia-Pacific countries were the major contributors, with more than 75% of the reported interceptions. Predictably, Australia, as the closest to New Zealand and the main destination and source for tourists and trade, appears to be, by far, the most common source of interceptions, accounting for 25% of the total interceptions of known origin. Since *Cx. pervigilans* is endemic, it is likely that specimens intercepted are from a local source. Japan was the second source of interceptions of a known origin, with 11.7%, followed by Ecuador, the major source of fruit imports (mainly bananas) to New Zealand.

**Entrance pathway:** The major pathway of entrance for mosquito interceptions to date has been by sea. More than 66% of known interceptions were at six New Zealand seaports (Table 3). Of these, 83% were at Ports of Auckland. The remaining 34% were in seven airports. Auckland International Airport was by far the main entry point, accounting for more than 81% of air pathway interceptions. Auckland was the main city of entry, with 201 records (134 Ports of Auckland and 67 Auckland International Airport), accounting for more than 82.5% of all interceptions (Table 3).

**Stages of development:** About 73% of the mosquito interceptions were recorded as adults (Table 4). The major pathway for both adults and larvae has been by sea, 58% and 88%, respectively (Table 4). *Ae. aegypti, Ae. Albopictus*, and *Cx. quinquefasciatus* made up 56% of the intercepted larvae (Table 1). Most larvae (48/66) originated from the South Pacific (29/66) and Asia (19/66) (Table 2). Used tyres and machinery accounted for about 60% of all larval interceptions and about 91% of larval interceptions with known modes of transport (Table 5).

**Means of transport:** This was unknown in 99 cases (≈41%) (Table 5). These were intercepted at seaports, airports, and their transitional facilities during mosquito port inspection and associated surveillance. Mosquitoes entered by ship in fresh fruit and vegetable containers (30/244), used machinery (29/244), and used tyres (25/244). Mosquitoes entered by airfreight in fruit and vegetable containers, personal luggage, and unspecified good containers, collectively accounting for 11% (27/244) of the cases.

**Years of interception**: Between 2001 and 2015, the records of mosquito interceptions in New Zealand varied between 6 and 21 records per year. However, an increase to 30 and 36 interceptions was recorded in 2016 and 2017, respectively (Figure 3). Overall, there was a significant mean annual increase of 7% (mean estimate 1.07, CI 1.03–1.12; *p* = 0.0009).

***Ae. aegypti* and *Ae. albopictus*:** To date, *Ae. aegypti* and *Ae. albopictus* were only intercepted in Auckland. Most *Ae. aegypti* were intercepted as adults arriving by aircraft. In contrast, most *Ae. albopictus* entered the country as larvae by sea (Table 6). *Ae. aegypti* was intercepted 19 times at Auckland International Airport and 10 times at Ports of Auckland, while *Ae. albopictus* was intercepted 19 times at Ports of Auckland and only once at Auckland International Airport. Most *Ae. albopictus* arrived in used machinery (40%) and used tyres (20%).

*Ae. aegypti* arrived mainly in used tyres and machinery (60%) (Table 6). While the country of origin of more than half of the *Ae. aegypti* intercepted was unknown (51.7%), Japan was the most common source of *Ae. albopictus* (40%) (Table 6).

For *Ae. Aegypti*, an annual increase of 20% (mean estimate 1.20, CI 1.08–1.34) was significant (*p* = 0.0009) and for *Ae. Albopictus*, there was no significant change (mean estimate 0.93, CI 0.84–1.03; *p* = 0.15). We note that there was considerable yearly variation and the main increase in *Ae. aegypti* occurred over the last four years (Figure 3). The monthly interception records of both species peaked in summer between December and February (Figure 4). This indicates a significant seasonality for *Ae. aegypti* (*p* = 0.003) and for *Ae. albopictus* (*p* = 0.047).

Since exotic mosquitoes in general, and *Ae. albopictus* and *Ae. aegypti* in particular, mainly arrived in used tyres and machinery, we also examined trends in import data for used tyres and machinery to compare with the interception data.

### 3.2. Trade Data Imports and International Flights

**Sea and air freight:** Between 2001 and 2017, the total gross weight of New Zealand’s sea and air freight imports increased by 60% (Appendix A) and 14% (Appendix A), respectively.

**International passenger flight arrivals:** The total number of international flights to New Zealand during the period of the study increased by more than 71% (from 22,180 to 38,027) (Appendix A). The most frequent international flights to New Zealand during the study period were from Sydney (146,082), Melbourne (78,497), and Brisbane (75,322). There was a ten-fold increase in the number of flights from Coolangatta airport (a tourist destination, Queensland Gold Coast) over the 17 year period. This makes Coolangatta the fourth highest source of international flights to New Zealand from airports where *Ae. aegypti* was established (after Brisbane, Los Angeles and Nadi) (Appendix A).

**Used tyres:** From 2001–2017, New Zealand imported about 5.5 million used tyres from 35 countries (Appendix A). About five million (≈91%) were from 21 countries where *Ae. albopictus* is established, and only about 300,000 (≈5%) were from 16 countries with *Ae. aegypti*. Japan, where *Ae. albopictus* is indigenous, supplied about 88% of the total number of used tyres. Although the total gross weight of New Zealand’s sea imports increased by more than 60% (Appendix A), there was an 86% decline in the quantity of imported used tyres from its peak in 2003 (from around 607,000 to 85,000). During 2001–2003, New Zealand’s annual used tyre imports averaged more than a half of a million. However, in 2017, this number significantly declined to about 85,000 (Appendix A) [45].

**Used vehicles:** In the past 17 years, New Zealand has imported a total of about 2.3 million used vehicles from 107 countries. Japan was by far the largest source, with more than 1.95 million used vehicles, representing about 88% of the total vehicle imports (Appendix A). New Zealand imported around 133,000 used vehicles per year from 2001 to 2017. However, concurrently with the global financial crisis, the average number of vehicle imports dropped to approximately 70,000 vehicles per year between 2007 and 2012 (Appendix A).

## 4. Discussion

### 4.1. Main Findings

Our findings add to earlier observations that *Aedes* mosquito vectors, especially *Ae. albopictus*, have extended their range via international travel and trade, especially via used tyres and machinery [5,6,7]. To date, *Ae. aegypti* and *Ae. albopictus* have been the two most commonly intercepted foreign vector mosquitoes. The Ports of Auckland and Auckland Airport were the main ports of arrival into New Zealand and the only points of entry for both *Ae. aegypti* and *Ae. albopictus*. Most of the interceptions probably originated from Japan or arrived via Australia. Japan, where *Ae. albopictus* is indigenous, was by far New Zealand’s largest supplier of both used tyres and vehicles and was also the largest source of *Ae. albopictus*.

The majority of *Ae. albopictus* were transported as larvae via shipping vessels, but the majority of *Ae. aegypti* were transported by air as adults. The latter is consistent with the increasing number of international flights to New Zealand from *Ae. aegypti* endemic areas, notably Queensland. Most of the exotic interceptions arrived by sea in used machinery and tyres. 

Most *Ae. albopictus* interceptions have been larvae, imported by sea, and most likely originating from Japan. The majority of *Ae. aegypti* have been intercepted as adults transported by air, potentially from neighbouring countries with direct and frequent flights to New Zealand. These findings are consistent with the biological features of *Ae. aegypti* and *Ae. Albopictus*, whose eggs are deposited in natural and artificial habitats where water levels fluctuate [54]. These eggs are stimulated to hatch by rising water levels, often many months later [21]. Larvae of many container-breeding mosquitoes are also able to survive food scarcity for several weeks, or even months, longer than any immature stages of other mosquito species (e.g., ground water mosquitoes) [55]. With suboptimal food, larvae of *Ae. albopictus* can survive between 58 and 80 days [56]. However, with enough food, they may develop to adults within two weeks. Another physiological adaptation of *Ae. albopictus* larvae that facilitates their survival in tyres is their superior tolerance to contaminants in tyre leachate compared with other mosquito species [57]. Furthermore, *Ae. albopictus* eggs are thermal and desiccation tolerant and may remain viable for several months [58].

Sailing time from Japan to Auckland, which is New Zealand’s main gateway for international trade and the main city of entry for foreign mosquitoes, is between 10–12.5 days [59]. The journey time is between 12–17 days from Munich [60], 15 days from San Francisco and three days from Australia [59,61]. Therefore, *Ae. albopictus* can survive a journey from Japan, Germany, or USA to New Zealand in larval and egg stages, possibly under extreme weather and food shortage conditions. The adults, however, cannot survive starvation for more than seven days [62]. It is, therefore, possible for a female mosquito locked in a vehicle to survive a journey from Australia, but this is unlikely to occur from more distant countries like the USA, Germany, and Japan [61].

It is clear that both *Aedes* species can travel to New Zealand via shipping and airplanes. However, the number of interceptions is relatively small (29 *Ae. aegypti* and 20 *Ae. albopictus* in 17 years). On the other hand, there were 39 detections of *Ae. aegypti* at Perth International Airport, Australia alone in a six month period (October 2015–April 2016) [63]. In the year ending June 2018, more than 10 million international passengers passed though Auckland Airport [64]. In the same period, about 4.4 million international passengers travelled through Perth International Airport [65]. Both airports receive frequent direct flights from *Ae. aegypti* endemic countries, such as Singapore, Indonesia, Thailand, and many Pacific Islands [64,65]. This comparison provides some support for the effectiveness of current New Zealand pre-border biosecurity measures to reduce mosquito introductions and prevent their establishment (Appendix A).

### 4.2. Current Risk Status for Arboviral Infections 

The New Zealand population is considered to have the highest per capita rate of international travel in the world, especially within the Asia Pacific region [66,67], where epidemics of arboviral disease are an ongoing feature. Imported cases of arboviral infections are reported every year among travellers to New Zealand arriving from endemic or epidemic regions, mainly Pacific Islands and Australia [68]. Travel and trade conditions, combined with the presence of exotic mosquito vectors and global climate change, increase the risk of the local transmission of mosquito-borne diseases in New Zealand [66]. A warmer climate will also facilitate latitudinal and altitudinal range expansion [38,42,69]. This situation is compounded by the recent unaccredited cruise ships that have arrived on New Zealand shores without being checked by the Ministry for Primary Industries [70]. This concern is exemplified by the finding of *Cx sitiens* larvae by Biosecurity NZ near the Kaipara in March 2018 [71]. *Cx. sitiens* is widespread in the Pacific, Australia and Southeast Asia and is a competent vector for several arboviruses, such as RRV [72].

Furthermore, according to a recent study carried out in Australia, intercepted *Ae. aegypti* mosquitoes detected at international ports in New Zealand and Australia had point mutations that confer strong resistance to synthetic pyrethroids, the only insecticide class used for aircraft disinsection validated by the World Health Organization [63]. 

### 4.3. Climatic Suitability for Aedes Mosquitos’ Establishment in New Zealand

Establishment and dispersal of exotic mosquitoes after introduction is only possible under suitable climatic conditions [34]. According to de Wet et al. [69], and under the current temperature and rainfall conditions, Auckland and the Northland regions are the most suitable areas in New Zealand for the establishment of *Ae. albopictus* [73]. However, projected increases in temperature, rainfall, and humidity could make these areas suitable for *Ae. aegypti* and extend the geographic distribution of *Ae. albopictus* to the south [69,73].

Summer 2017–2018 was New Zealand’s hottest summer on record across all regions, with average temperature of 18.8 °C, (2.1 °C above the 1981–2010 average). Auckland recorded the highest temperatures in the country [74]. Using a mid-range climate projection, Pearce et al. [75] reported that Auckland’s temperature is expected to increase by 0.5 to 4.2 °C by 2040–2110, depending on future concentrations of greenhouse gases in the atmosphere [75].

*Ae. aegypti* and *Ae. albopictus,* the most commonly intercepted “unwanted” species, were only intercepted in Auckland. Moreover, the majority of used tyres and machinery enters New Zealand at the Ports of Auckland. Arriving exotic mosquitoes in general, and *Ae. albopictus* in particular, are likely to find a suitable habitat around the ports of entry, while climate warming will facilitate their establishment and spread [76]. This local situation illustrates a global concern and one of the future human costs of globalising travel and trade.

### 4.4. Implications and Recommendations

The first line of defence against climatic influences on the establishment and spread of arbovirus diseases should be to prevent the entry of exotic mosquito vectors. This approach is likely to be far more cost-effective than attempting to contain and eradicate exotic mosquitoes after their establishment in New Zealand. We recommend a particular focus on biosecurity practices as follows:Regularly review mosquito interception practices as part of an integrated vector-borne disease surveillance system. Consider a surveillance sector approach [77] and advice from targeted research –based surveillance to identify potential improvements. Such surveillance will be essential to anticipate projected climatic influencesAs a component of regular reviews of mosquito surveillance and interception responses, there should be particular attention given to mosquito surveillance at major ports of entry, notably at the Ports of Auckland and Auckland Airport. As we recognise that interception data are, at least partially, relative to effort, this review should include improved standardisation of port surveillance procedures, recording, and schedules (e.g., install permanent traps and yearlong rather than seasonal operation) to provide a reliable baseline for future evaluation. This was a key limitation in the quality of data available for this present analysis and review.Increase the use of molecular methods to enhance mosquito interception surveillance. Effective biosecurity surveillance of mosquitoes will depend on having a high level of confidence in identifying mosquito species and origins. For example, only a third of interceptions linked to aircraft have a specified port of origin. Molecular identification (e.g., [78,79]) of unknown specimens and genetic origin analysis for unknown sources (e.g., [80]) are currently undertaken by an Australian laboratory [63]. It is essential to facilitate direct access to New Zealand and international molecular expertise and global reference material for this purpose.In addition to existing biosecurity practices, new regulations and requirements should be adopted for the discarding of waste tyres where they are within the 1600 meter-zone around ports (the dispersal distance of *Aedes* mosquitoes [81]).Review aircraft disinsection procedures for New Zealand. This review should pay particular attention to Auckland Airport and the recent increase in interceptions of *Ae. aegypti* [82] in view of the resistance to the pyrethroid pesticides [63] identified in specimens intercepted at New Zealand and Australian ports [63].Use research-based surveillance to regularly evaluate effectiveness and identify any specific gaps with regard to current biosecurity measures.

## 5. Conclusions

Continuing introductions of *Ae. aegypti* and *Ae. albopictus*, and their potential establishment, raise concerns about the initiation of locally transmitted mosquito-borne diseases in New Zealand in general and in the Auckland region in particular. In 1998, when Italy and Albania were the only European countries colonized by *Ae. albopictus*, Reiter [23] stated that “there is no reason to believe that the European countries will be more successful than the United States in blocking the importation of cargos infested with *Ae. albopictus*. In short, it seems we must accept the establishment of exotic species as an inevitable consequence of modern transportation technology”. *Ae. albopictus* has now been reported in at least 27 countries in Europe. This expansion was facilitated by human activities, in particular the trade of used tyres [22]. If New Zealand can learn any lesson from this experience with invasive mosquitoes, it is that there should be no complacency. Continued vigilance and investment in port surveillance are well justified by existing concerns, while future risks will be exacerbated by climatic factors.

## Figures and Tables

**Figure 1 tropicalmed-04-00101-f001:**
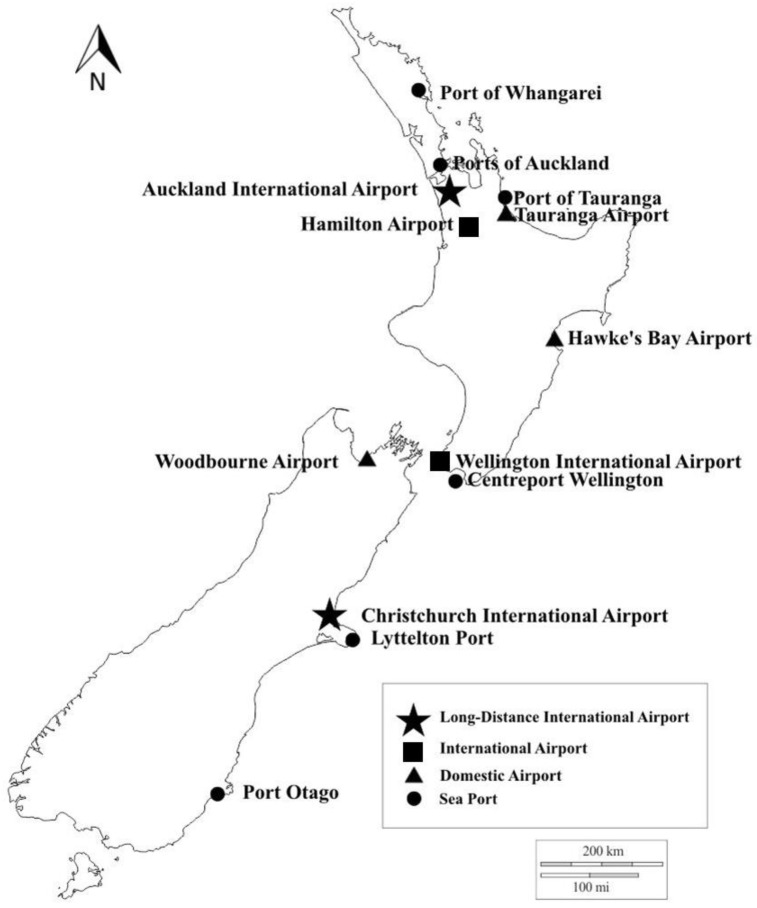
Ports of entry of mosquito interceptions in New Zealand, July 2001–March 2018.

**Figure 2 tropicalmed-04-00101-f002:**
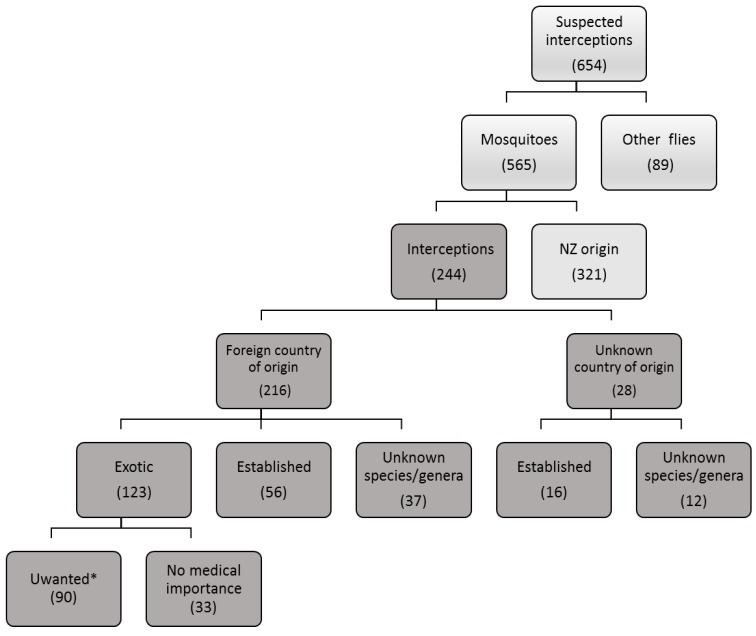
New Zealand Interception records, July 2001–March 2018.

**Figure 3 tropicalmed-04-00101-f003:**
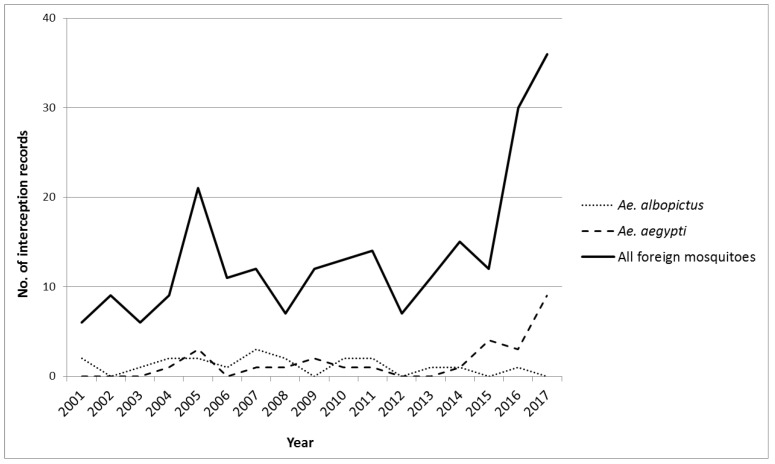
*Ae. aegypti*, *Ae. Albopictus*, and total foreign mosquito interception records in New Zealand, July 2001–December 2017.

**Figure 4 tropicalmed-04-00101-f004:**
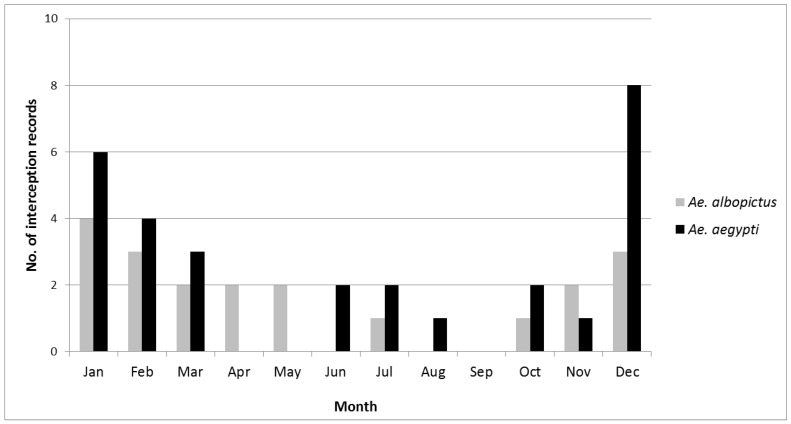
Monthly records of *Ae. aegypti* and *Ae. albopictus* intercepted in New Zealand, July 2001–December 2017.

**Table 1 tropicalmed-04-00101-t001:** Species and development stages of mosquito interception events in New Zealand, July 2001–March 2018. * On Ministry of Health “Unwanted Organisms Register”.

Species	No. of Events (Larvae)
***Culex***	**107 (14)**
*Cx. quinquefasciatus*	60 (11)
*Cx. sitiens **	8 (1)
*Cx. australicus*	6 (0)
*Cx. pervigilans*	5 (0)
*Cx. pipiens **	2 (0)
*Cx. gelidus **	2 (0)
*Cx. nigripalpus*	1 (0)
*Cx. annulirostris **	1(0)
*Cx. ocossa*	1 (0)
*Cx. tritaeniorhynchus*	1 (0)
*Cx. fuscocephala*	1 (0)
*Cx.* sp.	19 (2)
***Aedes***	**108 (47)**
*Ae aegypti **	29 (11)
*Ae. albopictus **	20 (15)
*Ae. notoscriptus*	14 (6)
*Ae. vexans*	8 (0)
*Ae. camptorhynchus **	6 (3)
*Ae. japonicus **	6 (4)
*Ae. vigilax **	6 (1)
*Ae. taeniorhynchus*	3 (0)
*Ae. polynesiensis **	2 (2)
*Ae. vittiger*	2 (0)
*Ae. alternans*	2 (0)
*Ae. tremulus*	1 (1)
*Ae. cinereus*	1 (0)
*Ae. infirmatus*	1 (0)
*Ae. sollicitans*	1 (0)
*Ae. cooki*	1 (1)
*Ae. togoi **	1 (1)
*Ae. sierrensis **	1 (1)
*Ae.* sp.	3 (1)
***Anopheles***	**6 (1)**
*A. siniensis **	1 (0)
*A. subpictus **	1 (0)
*A. stephensi **	1 (0)
*A. crucians **	1 (0)
*A. culicifacies **	1 (1)
*A. albimanus **	1 (0)
**Other**	**9 (3)**
*Mansonia humeralis*	1 (0)
*Culiseta annulata*	1 (0)
*Coquillettidia nigricans*	1 (0)
*Uranotaenia* sp.	1 (0)
*Verralina funerea*	1 (0)
*Mansonia titillans*	1 (0)
*Toxorhynchites speciosus*	1 (1)
*Tripteroides bambusa*	1 (1)
*Uranotaenia novobscura*	1 (1)
***Mosquito* spp.**	**14 (1)**
**Total**	**244 (66)**

**Table 2 tropicalmed-04-00101-t002:** Countries of origin and development stages of mosquito interception events in New Zealand, July 2001–March 2018.

Origin of Transport	No. Events (Larvae)	% by Total
**South Pacific**	**95 (29)**	**38.9**
Australia	48 (9)	19.7
Fiji	12 (0)	4.9
Vanuatu	7 (6)	2.9
Samoa	6 (4)	2.5
New Caledonia	6 (0)	2.5
Cook Islands	5 (5)	2.0
Tonga	5 (0)	2.0
Wallis and Futuna	2 (2)	0.8
Niue	1 (1)	0.4
Papua New Guinea	1 (1)	0.4
Guam	1 (0)	0.4
Noumea	1 (0)	0.4
**Asia**	**52 (19)**	**21.3**
Japan	23 (15)	9.4
India	4 (1)	1.6
Philippines	4 (0)	1.6
China	4 (0)	1.6
Thailand	3 (0)	1.2
Hong Kong	3 (0)	1.2
Malaysia	2 (1)	0.8
Korea	2 (1)	0.8
Taiwan	2 (0)	0.8
Vietnam	2 (0)	0.8
Singapore	2 (1)	0.8
Cambodia	1 (0)	0.4
**Americas**	**46 (4)**	**18.9**
Ecuador	21 (0)	8.6
USA	17 (4)	7.0
Chile	3 (0)	1.2
Canada	2 (0)	0.8
Panama	1 (0)	0.4
Argentina	1 (0)	0.4
Colombia	1 (0)	0.4
**Europe**	**4 (0)**	**1.6**
Netherlands	2 (0)	0.8
Germany	2 (0)	0.8
**Unknown**	**47 (15)**	**19.3**
**Total**	**244 (66)**	**100**

**Table 3 tropicalmed-04-00101-t003:** Entrance pathway and development stages of mosquito interception events in New Zealand, July 2001–March 2018.

Pathway	Port	No. of Events (Larvae)	% by Entrance Pathway	% by Total
**By sea**	Ports of Auckland	134 (52)	83.2	54.9
	Lyttelton Port	11 (5)	6.8	4.5
	CentrePort Wellington	7 (1)	4.4	2.9
	Port of Tauranga	7 (0)	4.4	2.9
	Port of Whangarei	1 (0)	0.6	0.4
	Port Otago	1 (0)	0.6	0.4
**Total interceptions by sea**		**161 (58)**	**100**	**66**
**By air**	Auckland International Airport	67 (6)	80.7	27.5
	Christchurch International Airport	9 (0)	10.8	4.4
	Wellington Airport	3 (0)	3.6	1.2
	Tauranga Airport	1 (0)	1.2	0.4
	Hamilton Airport	1 (1)	1.2	0.4
	Hastings airport	1 (0)	1.2	0.4
	Marlborough Airport	1 (1)	1.2	0.4
**Total interceptions by air**		**83 (8)**	**100**	**34**
**Total interceptions**		**244 (66)**	**100**	**100**

**Table 4 tropicalmed-04-00101-t004:** Development stages and pathways of mosquito interception events in New Zealand, July 2001–March 2018.

Stage	Entrance Pathway	No. of Events	% by Stage	% by Total
**Adult**	By air	75	42	31
	By sea	103	58	42
**Total adult interceptions**		**178**	**100**	**73**
**Larvae**	By air	8	12	3
	By sea	58	88	24
**Total larvae interceptions**		**66**	**100**	**27**
**Total interceptions**		**244**	**100**	**100**

**Table 5 tropicalmed-04-00101-t005:** Mosquito interception events in New Zealand, July 2001–March 2018.

Pathway	Mean of Invasion		No. Events (Larvae)	% by Pathway	% by Total
**By air**	Air Containers	Roses, fresh fruits, and vegetables	11 (0)	13.3	4.5
		Unspecified	9 (0)	10.8	3.7
	Luggage		7 (1)	8.4	2.9
	Unknown	Inspection at and around ports/transitional facilities	35 (1)	42.2	14.3
		Aircrafts inspection	14 (2)	16.9	5.7
		Surveillance traps	7 (4)	8.4	2.9
**Total interceptions by air**			**83 (8)**	**100**	**34**
**By sea**	Used tyres		25 (19)	15.5	10.2
	Used machinery		29 (20)	18	11.9
	Containers	Fresh fruits and vegetables	30 (1)	18.6	12.3
		Manufactured goods	9 (0)	5.6	3.7
		Empty	5 (0)	3.1	2
		Unspecified	20 (2)	12.4	8.2
	Unknown	Ports/ transitional facilities inspection	14 (3)	8.7	5.7
		On ships- inspection	25 (12)	15.5	10.2
		Surveillance traps	4 (1)	2.5	1.6
**Total interceptions by sea**			**161 (58)**	**100**	**66**
**Total interceptions**			**244 (66)**	**100**	**100**

**Table 6 tropicalmed-04-00101-t006:** Origin of *Ae. aegypti* and *Ae. albopictus* intercepted in New Zealand, July 2001–March 2018.

Species (No.)	Location of Interception (No.)	Stage (No.)	Mode of Entry (No.)	Origin of Transport (No.)
***Ae. aegypti* (29)**	**Auckland International Airport (19)**	**Adult (15)** **Larvae (4)**	**Unknown (17), Fruit container (1), Luggage (1)**	Unknown (12), USA (1), New Caledonia (1), Japan (1), Australia (1), Philippines (1), Fiji (1), Cambodia (1)
	Ports of Auckland (10)	Adult (3) Larvae (7)	Used machinery (4), Used tyres (2), Unknown (2), Empty Container (1), Unspecified good container (1)	Unknown (3), Cook Islands (2), Tonga (1), Samoa (1), Papua New Guinea (1), Vanuatu (1), Futuna (1)
***Ae. albopictus* (20)**	Auckland International Airport (1)	Adult (1)	Luggage (1)	Taiwan (1)
	Ports of Auckland (17) and their transitional facility (2)	Adult (4) Larvae (15)	Used machinery (8), Used tyres (4), Unknown (6), Unspecified good Container (1)	Japan (8), Vanuatu (3), USA (2), Unknown (2), Cook Islands (1), Malaysia (1), Korea (1), Vietnam (1)

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
