# Peer review of "Intercepted Mosquitoes at New Zealand’s Ports of Entry, 2001 to 2018: Current Status and Future Concerns"

_tropicalmed, 2019, doi:10.3390/tropicalmed4030101_

Round 1

Reviewer 1 Report

Overall this is an interesting summary of an invasive biosecurity threat, so far contained at NZ ports of entry. It might be a valuable, readable paper if subject to some drastic editing.

The document needs checked for the veracity of some statements (see comments below) and some of the wording is bit odd – there are quite a few definite articles (“the”) missing.

Although the general topic is topical, much of the data presented is of very specific interest to a small number of readers and should be relegated to supplementary materials. Tables and graphs could be rationalised accordingly

Much of the text needs clarification – the immediate meaning is not clear. The manuscript could do with someone reading it carefully and doing some editing.

Specific comments:

In the abstract, surely, it’s less tropically adapted rather than “less cool-adapted” mosquitoes that are most at risk of invasion and establishment. New Zealand is already suitable for temperate species invasions and, despite climate change, it’s still marginal for anything else. If I’m wrong about that, please address and expand in manuscript.

Interesting that the abstract identifies most incursions as coming from Australia and Japan – perhaps room here to stress impact of Australian biosecurity failures on NZ risks. 

Line 46: Reference recent PLoS One publication re: Ae aegypti invasions and water tanks (Trewin et al).

Line 40. Reference to figure is out of place.

Line 63. Highly unlikely that Ae vigilax has been involved in dengue transmission in Fiji

Line 66.  Tokyo is humid sub-tropical not temperate. Be specific about the mosquito that caused the outbreak (Ae albopictus)

Lines 71 onwards. I’m uncomfortable with direct association between climate change and arbovirus transmission - there are many factors that mitigate the risk, particularly around human behaviours. Also, the projections given are outdated. Climate Change Projections for New Zealand suggest 0.2–1.7°C by 2040, 0.1–4.6°C by 2090, and 0.3–5.0°C by 2110.

Line 128. Add detail on trapping effort and types of traps used plus consideration of whether these are adequate. If you’re not looking properly, or using optimal tools, you won’t catch much.

Line 158. Not clear to me what the majority of the “suspected interventions’ were – please add detail. It seems like the NZ definition of interception is of any insect whether exotic or endemic?  

Line 167 – If Ae camptorhynchus is considered eradicated, probably should not be referred to as established.

Table 1 and Table 2. Do the numbers refer to total individual interceptions (larvae plus adults) and then larval interceptions in brackets? Are those figures numbers of interception events or numbers of individuals? The tables don’t tell you whether Australia and Japan are sources of the riskiest invaders Ae aegypti and Ae albopictus. Are they?

Line 214 onwards – I don’t understand this paragraph. Does the annual increase refer to year on year, or just 2016-2017? Does it include native Cx quinquefasciatus? If so why not just analyses those species that are not resident?

Line 230 - what are the methods comparing time trends that resulted in the p=0.0005 value? Are the annual increases year on year? Or just a subset? Or between 2001 and 2018?

Line 250 onwards: The trade data is without context. Make something meaningful out of it and put the detail in supplementary.

Line 275-286. I’m sure some of these graphs and tables can be combined / relegate to supplementary.

Line 299. NZ does have some flights to Cairns – the only Ae aegypti endemic destination in Australia - during New Zealand's autumn and winter. How does the timing of these correlate with detections? Does it suggest that NZs disinsection is not working?

Line 318. Odd emphasis on zinc when there are so many other toxins leaching from tyres.

Line 287. The discussion, recommendations and conclusions are mostly waffle. To maintain the interest of the reader, I suggest that they are cut drastically –figure out some main messages and keep the surrounding text relevant. Lines 321- 328 for example are mostly unnecessary detail.

Some of the key points at the end have no relationship to the preceeding text:

Points made repetitively on lines 416, 427 and 430: Yes, an adequately resourced and expert biosecurity service is required but this paper tells us nothing about that (not even detailing the intensity of the trapping effort or the types of traps used).

Line 421: molecular phylogenetic won’t tell you much unless you have access to a global database of specimens. Do they? What’s required to set one up?

Author Response

We are grateful for the effort and constructive (and sometimes challenging!) comments provided by the reviewers. Please note that attached are the supplementary materials and two copies of the revised manuscript; a clean one with our amendments highlighted in red (reviewer 1) and green (reviewer 2) and another one with the track changes. As it is only allowed to upload one file, both documents are combined and uploaded. The order in combined document is as follows: 1st the clean manuscript, 2nd the manuscript with the track changes and 3rd the supplementary materials.

Point 1: The document needs checked for the veracity of some statements (see comments below) and some of the wording is bit odd – there are quite a few definite articles (“the”) missing.

Although the general topic is topical, much of the data presented is of very specific interest to a small number of readers and should be relegated to supplementary materials. Tables and graphs could be rationalised accordingly

Much of the text needs clarification – the immediate meaning is not clear. The manuscript could do with someone reading it carefully and doing some editing

Response 1: The whole paper has been extensively revised for clarity and readability as well as removing odd wording. Many sections, figures and tables have been moved to supplementary materials (see attached manuscript with the track changes).  Further, the layout of tables has been changed to be better in keeping with the formatting requirements of the journal. 

Point 2: In the abstract, surely, it is less tropically adapted rather than “less cool-adapted” mosquitoes that are most at risk of invasion and establishment. New Zealand is already suitable for temperate species invasions and, despite climate change; it is still marginal for anything else. If I am wrong about that, please address and expand in manuscript.

Response 2: This is our slipup. We have amended the wording as follows: “Climate change makes New Zealand, in particular the northern regions, an increasingly suitable environment for tropically adapted exotic mosquito vectors to become established.”

Point 3: Interesting that the abstract identifies most incursions as coming from Australia and Japan – perhaps room here to stress impact of Australian biosecurity failures on NZ risks.

Response 3: Australian and Japanese  biosecurity agencies are likely to be concerned only with incoming organisms & vessels.   However, outgoing transport is usually subject to inspection by NZ offshore Biosecurity agencies.  We have expanded recommendation #6 to specifically mention offshore agencies & practices.

Point 4: Line 46: Reference recent PLoS One publication re: Ae aegypti invasions and water tanks (Trewin et al).

Response 4 The recommended paper has been referenced. The wording now reads as follows: “Unlike Ae albopictus, Ae aegypti does not overwinter in the egg stage. Instead, the active stages minimize exposure to unsuitable environmental conditions by utilizing sheltered sites in urban settings (e.g. water tank) [19].”

Point 5: Line 40. Reference to figure is out of place.

Response 5: The word “figure” has been deleted.

Point 6: Line 63. Highly unlikely that Ae vigilax has been involved in dengue transmission in Fiji.

Response 6: We reconsidered information regarding the habits of Ae vigilax and have amended the text as follows:

Further, dengue and Ross River Virus (RRV) infection outbreaks followed the introduction of Ae. albopictus into the Solomon Islands [1], Ae vigilax into Fiji [31] , respectively, and most of the Western Pacific region by aircraft [1, 31].

Point 7: Line 66.  Tokyo is humid sub-tropical not temperate. Be specific about the mosquito that caused the outbreak (Ae albopictus).

Response 7: Regarding the first part of the comment, according to Japan Meteorological Agency, Japan has “a climate ranging from subarctic in the north to subtropical in the south”. Accordingly, the wording reads now as follows: “Despite the predominate temperate climate (subarctic in the north and subtropical in the south) [32], Ae. albopictus is prevalent in Japan, with northward expansion (annual mean temperature of 11°C) [33]. Tokyo recorded an outbreak of dengue in early autumn 2014 for the first time since World War II [34]. Dengue epidemics in Japan are likely to increase over the next decades, exacerbated by the continuing geographic expansion of Ae. albopictus vectors and the conducive climatic conditions [35].

Regarding to the 2nd part of the comment, in line 69, “Aedes” has been specified as “Ae. Albopictus”.

Point 8: Lines 71 onwards. I’m uncomfortable with direct association between climate change and arbovirus transmission - there are many factors that mitigate the risk, particularly around human behaviours. Also, the projections given are outdated. Climate Change Projections for New Zealand suggest 0.2–1.7°C by 2040, 0.1–4.6°C by 2090, and 0.3–5.0°C by 2110.

Response 8: Regarding the comment on the association between climate change and arbovirus transmission, the wording has been amended and now reads as follows: “Increasing temperature has been also implicated as a major factor in the establishment and re-establishment of Aedes species and their associated diseases in Europe [4, 32].”

Regarding the second part of the comment on the projections, the projections given in line 73 are the global projections according to “Intergovernmental Panel on Climate Change” report in 2018, “Global warming is likely to reach 1.5°C between 2030 and 2052 if it continues to increase at the current rate”. Accordingly, the wording has been changed as follow:

“Global climate models project a rise in mean temperatures of 1.5°C by the year between 2030 and 2052 [39]”.

Point 9: Line 128. Add detail on trapping effort and types of traps used plus consideration of whether these are adequate. If you’re not looking properly, or using optimal tools, you won’t catch much.

Response 9: Types of traps have been added. The paragraph now reads as follows:

“The surveillance consists of routine monitoring surveys including adult (CO2 baited light traps) and larvae (WHO standard tyre traps) trapping, larval surveys (WHO standard dipping method) and interception responses.”

Regarding the efforts, all the surveillance involved use of WHO standard adult and larval capture methods. However, we are aware that most biosecurity effort is concentrated at sites deemed to be high risk (e.g. Auckland and international airports and seaports) & and at some other sites at high risk times, and that there is a need to ensure greater standardisation of practices and procedures.  This should be a priority for future biosecurity reviews (recommendation# 4).

Point 10: Line 158. Not clear to me what the majority of the “suspected interventions’ were – please add detail. It seems like the NZ definition of interception is of any insect whether exotic or endemic? 

Responds 10: “Suspected interception” is an interim category, which may include non-mosquito flies or local mosquito species, while an “interception event” is the confirmed detection of an exotic mosquito species on a particular date [refer line 107-116 (in the original manuscript) and Fig 2).

Point 11: Line 167 – If Ae camptorhynchus is considered eradicated, probably should not be referred to as established.

Responds 11:    This species became established in the late 1990s and was declared to be eradicated in 2010 (Kay and Russell 2013) hence was ‘established’ for part of the review period. Further, the sentence that follows, addresses this concern, “Although Ae. camptorhynchus was declared to be eradicated from New Zealand in June 2010, it remains a high-risk species on the register [52]”.

Point 12: Table 1 and Table 2. Do the numbers refer to total individual interceptions (larvae plus adults) and then larval interceptions in brackets? Are those figures numbers of interception events or numbers of individuals? The tables do not tell you whether Australia and Japan are sources of the riskiest invaders Ae aegypti and Ae albopictus. Are they?

Responds 12: The first number refers to number of interception events and includes adults and/or larvae, while the number in the brackets refers to number of those events with larvae. Captions for Tables 1-3 have been modified to clarify this.

Point 13: Line 214 onwards – I don’t understand this paragraph. Does the annual increase refer to year on year, or just 2016-2017? Does it include native Cx quinquefasciatus? If so why not just analyses those species that are not resident?

Responds 13: Annual increase refers to year on year. Interceptions of Cx quinquefasciatus are assigned as “foreign” where they are associated with transport of foreign origin, mainly Australia. This assumes that resident Cx quinquefasciatus are not included.

 Point 14: Line 230 - what are the methods comparing time trends that resulted in the p=0.0005 value? Are the annual increases year on year? Or just a subset? Or between 2001 and 2018?

Response 14:  The annual increases are compared over the entire period 2001-18. Time trends in interception counts were modelled with negative binomial regression with overdispersion using the Genmod procedure in the statistical package SAS 9.4 (SAS Institute Inc., Cary, NC).

Point 15: Line 250 onwards: The trade data is without context. Make something meaningful out of it and put the detail in supplementary.

Recent reductions in interceptions of Ae. Albopictus larvae, and increases in Ae. Aegypti adults, are consistent with the reduction in both used tyres from Japan, and the increase in flight arrivals. We consider that this interpretation supports our recommendations.

Point 15: Line 275-286. I’m sure some of these graphs and tables can be combined / relegate to supplementary.

Response 16: As suggested, these have been transferred to supplementary materials.

Point 15: Line 299. NZ does have some flights to Cairns – the only Ae aegypti endemic destination in Australia - during New Zealand's autumn and winter. How does the timing of these correlate with detections? Does it suggest that NZs disinsection is not working?

Response 16: Interceptions assigned to Australian origin have no further specification. Data available suggest a summer peak December-March.

Point 17: Line 318. Odd emphasis on zinc when there are so many other toxins leaching from tyres.

Response 17: We have not specifically researched this topic although are aware that air breathing of larvae provides some tolerance to water contamination. The reference to zinc provided an example, but we agree that it is better to refer generally to the mix of contaminants in tyre leachate cited by these authors [52]. Accordingly, the text now reads as follows: “Another physiological adaptation of Ae. albopictus larvae that facilitates their survival in tyres is their superior tolerance to contaminants in tyre leachate compared with other mosquito species [56].”

Point 18: Line 287. The discussion, recommendations and conclusions are mostly waffle. To maintain the interest of the reader, I suggest that they are cut drastically –figure out some main messages and keep the surrounding text relevant. Lines 321- 328 for example are mostly unnecessary detail.

Response 18: These sections have been extensively revised for clarity and readability. The section of “National concerns and responses for mosquito interceptions” in the discussion was transferred to the supplementary materials. We referred to it in the last paragraph of the first part of the discussion (main findings).

Regarding lines 321-328 (in original manuscript); we believe that this is a key paragraph in the discussion. As the transportation origin of more than half of the Ae. aegypti intercepted was unknown, this leaves open the possibility that Australia may be the main source of Ae aegypti intercepted in  NZ.  This is supported by the information to date that most Ae aegypti were intercepted as adults arriving by air transport, and the consensus that this species does not survive prolonged travel as adults under aircraft conditions. In this regard Sydney, in particular, is a major hub for both travel to NZ and elsewhere.   While assessment period of the study is between 2001and 2018, this species was present until recent time along the eastern seaboard of Queensland down to Sydney, although is now eliminated from Brisbane & NSW (Trewin et al. 2017).

See: https://journals.plos.org/plosntds/article?id=10.1371/journal.pntd.0005848

Point 19: Some of the key points at the end have no relationship to the preceeding text: Points made repetitively on lines 416, 427 and 430: Yes, an adequately resourced and expert biosecurity service is required but this paper tells us nothing about that (not even detailing the intensity of the trapping effort or the types of traps used).

Response 19: We have revised the text for repetition. We hope that our findings justify the need for a robust and well-resourced biosecurity system. The structure of the recommendations has been modified for clarity.

Point 20: Line 421: molecular phylogenetic won’t tell you much unless you have access to a global database of specimens. Do they? What’s required to set one up?

Response 20: We recommended that, “It is essential to facilitate direct access to New Zealand and international molecular expertise and global reference material for this purpose”. Indeed, Part of the facilitation will be having an open access to global database of the specimens of interest. 

Reviewer 2 Report

Dear authors,

Your manuscript entitled: "Intercepted mosquitoes at New Zealand ports of entry, 2001 to 2018: current status and future concerns" is of big interest since it shows the problems concerning the entering of exotic mosquito species into New Zealand due to external factors, which, together with climate change, represents a threat to public health in that country. However, I must refer that this research is focused in New Zealand and, in this way, could possibly be submitted to a more regional scientific journal. Although the text is clear and support by enough data, there are some points I must refer and others that should be changed:

Introduction section:

Lines 33-34: When the authors mention the arboviral diseases that some mosquitoes can transmit, I think that should be referred the distribution of these diseases worldwide in a paragraph.

Line 35 and many others along the text: The authors must change Ae albopictus and Ae aegypti to Ae. albopictus and Ae. aegypti, putting a stop sign (.) after the abbreviation Ae

Line 40: Remove the word "figure" form the sentence.

Line 44-45: When the authors mention "(...) Madeira island, southwest of Portugal [15], (...)" are they referring to the localization of the island or Ae. aegypti as also established in the SW region of Portugal? It is not clear from the text.

Line 58 and other along the text: There is a space lacking between “aegypti and “[2]”.

Line 61 and others along the text: There is a full stop between “indigenous” and “[1]”.

Line 65: A stop sign (.) must be placed between “conditions” and “Despite”.

Lines 72-73: The rise of temperatures mentioned in this part of the text is the worst-case scenario? Or there is a rise in temperatures below 2 ºC pointed out for 2050 and below 4 ºC by the end of the 21st century in other studies for New Zealand? I think the authors must include here the results of 1 or 2 more scientific studies concerning global warming and the rise of temperatures for New Zealand or nearby.

Line 79: For the New Zealand 12 native mosquito species must be a reference in the text.

Line 89: Which two endemic species are potential arbovirus vectors? Name them on the text.

Materials and Methods section:

Line 110: Since it is the first time that it is mentioned on the text, MIP must be full written here and not in line 351.

Line 133: Only 7 of the 35 air and seaports where biosecurity surveillance is undertaken were analysed in this study. Why weren’t the others? Do they not receive persons or materials form other countries? Cannot these places be a zone of entrance of exotic mosquitoes? I think this point must be further explained.

Figure 1: In the label of the map, the small star that represents “International Airport” must be changed to another symbol, so it is not mistaken for “Long-Distance International Airport”.

Line 200: Ae albopictusand à the word “and” must not be in italic.

Table 5: There are some entrances that are not well described, like, by instance: “Ports/International”, “Inspection at and”, “Fresh fruits and”. This must be corrected.

Discussion section:

Line 300: The sentence should be changed as: “Used machinery and tyres, which are transported mainly via sea, were, together, the main means of entry (…)”

Line 336: Rewrite the sentence: “In the year ended June 2018 (…)”.

Line 356: At the end of the line, “Culex sitiens” must be written in italic.

Line 358: Put here the meaning of RRV.

Line 396: Choose between “began” or “initiated”.

Line 438: Remove the “m” after 1600, since it is written again in “meter-zone”.

Author Response

We are grateful for the effort and constructive (and sometimes challenging!) comments provided by the reviewers. Please note that attached are the supplementary materials and two copies of the revised manuscript; a clean one with our amendments highlighted in red (reviewer 1) and green (reviewer 2) and another one with the track changes. As it is only allowed to upload one file, both documents are combined and uploaded. The order in combined document is as follows: 1st the clean manuscript, 2nd the manuscript with the track changes and 3rd the supplementary materials.

We discussed whether this manuscript should be offered to a local journal; but felt that the climate concerns combined with the globalising of travel & trade as means of introducing  vector organisms & other pests should be highlighted  as an international issue, even for geographically isolated areas such as NZ.   Reverse seasonality has also provided some past protection from northern hemisphere pests but this is expected to become less of a protective factor in future conditions.

Note: The layout of tables has been changed to be better in keeping with the formatting requirements of the journal.

Introduction section:

Point 1: Lines 33-34: When the authors mention the arboviral diseases that some mosquitoes can transmit, I think that should be referred the distribution of these diseases worldwide in a paragraph.

Response 1: The following sentence has been added: “Despite the global distribution of these arboviruses, the majority are found in tropical and subtropical climate zones where Aedes mosquitos are prevalent. [9].

Point 2 Line 35 and many others along the text: The authors must change Ae albopictus and Ae aegypti to Ae. albopictus and Ae. aegypti, putting a stop sign (.) after the abbreviation Ae

Response 2: Changed as suggested.

Point 3: Line 40: Remove the word "figure" form the sentence.

Response 3: removed as suggested.

Point 4: Line 44-45: When the authors mention "(...) Madeira island, southwest of Portugal [15], (...)" are they referring to the localization of the island or Ae. aegypti as also established in the SW region of Portugal? It is not clear from the text.

Response 4: The wording has been amended to address this as follows: It has recently re-established in Madeira island (southwest of Portugal) [15], around the Black Sea [16], and in Netherlands [17].”

Point 5: Line 58 and other along the text: There is a space lacking between “aegypti and “[2]”.

Response 5: The space has been added.

Point 6: Line 61 and others along the text: There is a full stop between “indigenous” and “[1]”.

Response 6: the full stop has been deleted as suggested.

Point 7: Line 65: A stop sign (.) must be placed between “conditions” and “Despite”.

Response 7: The full stop has been added as suggested.

Point 8: Lines 72-73: The rise of temperatures mentioned in this part of the text is the worst-case scenario? Or there is a rise in temperatures below 2 ºC pointed out for 2050 and below 4 ºC by the end of the 21st century in other studies for New Zealand? I think the authors must include here the results of 1 or 2 more scientific studies concerning global warming and the rise of temperatures for New Zealand or nearby.

Response 8:  The wording has been changed to be “Global climate models project a rise in mean temperatures of 1.5°C between 2030 and 2052 [39].” (reviewer one also raised the same concern).

Reference [39]: Intergovernmental Panel on Climate Change, Global Warming of 1.5° C: An IPCC Special Report on the Impacts of Global Warming of 1.5° C Above Pre-industrial Levels and Related Global Greenhouse Gas Emission Pathways, in the Context of Strengthening the Global Response to the Threat of Climate Change, Sustainable Development, and Efforts to Eradicate Poverty. 2018: Intergovernmental Panel on Climate Change.

Point 9: Line 79: For the New Zealand 12 native mosquito species must be a reference in the text.

Response 9: The reference has been added as suggested.

Point 10: Line 89: Which two endemic species are potential arbovirus vectors? Name them on the text.

 Response 10: The two mosquitoes have been named in the text as follows:According to Kramer et al. [39], at least two endemic species (Cx. pervigilans and Ae. antipodeus) are also potential arbovirus vectors.”

Materials and Methods section:

Point 11: Line 110: Since it is the first time that it is mentioned on the text, MIP must be full written here and not in line 351.

Response 11:  The full words were spelled out as suggested.

Point 12: Line 133: Only 7 of the 35 air and seaports where biosecurity surveillance is undertaken were analysed in this study. Why weren’t the others? Do they not receive persons or materials from other countries? Cannot these places be a zone of entrance of exotic mosquitoes? I think this point must be further explained.

Response 12: The others are small and local/private ports.

Point 13: Figure 1: In the label of the map, the small star that represents “International Airport” must be changed to another symbol, so it is not mistaken for “Long-Distance International Airport”.

Response 13: The small stars have been replaced by squares.

Point 14: Line 200: Ae albopictus and  the word “and” must not be in italic.

Response 14: Fixed as suggested. 

Point 15: Table 5: There are some entrances that are not well described, like, by instance: “Ports/International”, “Inspection at and”, “Fresh fruits and”. This must be corrected.

Response 15: The words were incomplete because the rows were narrow. The rows were widened and the problem has been fixed. 

Discussion section:

Point 16: Line 300: The sentence should be changed as: “Used machinery and tyres, which are transported mainly via sea, were, together, the main means of entry (…)”

Response 16: Your suggestion has been taken.

Point 17: Line 336: Rewrite the sentence: “In the year ended June 2018 (…)”.

Response 17: The sentence has been rewritten as follows: “In the year ended June 2018, more than 10 million international passengers passed though Auckland Airport [64].

Point 18: Line 356: At the end of the line, “Culex sitiens” must be written in italic.

Response 18: The sp. was write in italic. Further, the “Culex” has been replaced by “Cx.” 

Point 19: Line 358: Put here the meaning of RRV.

Response 19: RRV was spilled out in the introduction (5th paragraph) after the other reviewer suggested some modifications.

Point 20: Line 396: Choose between “began” or “initiated”.

Response 20: The word “began” has been deleted as suggested.

Point 21: Line 438: Remove the “m” after 1600, since it is written again in “meter-zone”.

Response 21: The “m” has been removed as suggested.
